# The Risk Factors and Mechanisms of Azole Resistance of *Candida tropicalis* Blood Isolates in Thailand: A Retrospective Cohort Study

**DOI:** 10.3390/jof8100983

**Published:** 2022-09-20

**Authors:** Teera Leepattarakit, Orawan Tulyaprawat, Popchai Ngamskulrungroj

**Affiliations:** Department of Microbiology, Faculty of Medicine Siriraj Hospital, Mahidol University, Bangkok 10700, Thailand

**Keywords:** *Candida tropicalis*, azole resistance, risk factors

## Abstract

In recent decades, an epidemiological shift has been observed from *Candida* infections to non-*albicans* species and resistance to azoles. We investigated the associated factors and molecular mechanisms of azole-resistant blood isolates of *C. tropicalis*. Full-length sequencing of the *ERG11* gene and quantitative real-time RT-PCR for the *ERG11*, *MDR1*, and *CDR1* genes were performed. Male sex (odds ratio, 0.38), leukemia (odds ratio 3.15), and recent administration of azole (odds ratio 10.56) were associated with isolates resistant to azole. *ERG11* mutations were found in 83% of resistant isolates, with A395T as the most common mutation (53%). There were no statistically significant differences in the expression of the *ERG11*, *MDR1*, and *CDR1* genes between the groups resistant and susceptible to azole. The prevalence of azole-resistant isolates was higher than the usage of antifungal drugs, suggesting the possibility of environmental transmission in the healthcare setting. The unknown mechanism of the other 17% of the resistant isolates remains to be further investigated.

## 1. Introduction

*Candida* species are the most common cause of invasive fungal infections, affecting around three-quarters of a million patients per year [1]. Although *C. albicans* is the major species causing worldwide candidemia, the prevalence of non-*albicans* candidemia is increasing, especially in the Asia-Pacific region [2]. Moreover, *C. tropicalis* has become the major cause of *Candida* bloodstream infection in Thailand [3,4].

*C. tropicalis* bloodstream infection also raised concerns about the high rate of death from cases [4]. Furthermore, the prevalence of azole resistance is an increasing trend in Thailand, South Korea, and China, which might be associated with the use of fungicides in agriculture and cross-transmission to the hospital [5,6,7,8]. However, the azole resistance rate is lower in some other Asian countries including India and Iran [9,10]. The prevalence of fluconazole-resistant *C. tropicalis* isolates in Thailand increased from 20.2% in 2013–2015 to 37.8% in 2016–2017 [8,11]. Similarly, in China, fluconazole-resistant isolates increased from less than 10% in 2010–2012 to more than 30% in 2016–2017 [7]. Several studies reported the relationship between azole-resistant *Candida* infection and identified multiple risk factors including previous exposure to fluconazole and a history of bacterial bloodstream infection [12,13]. The description of *C. tropicalis* azole resistance was attributed to the four major mechanisms related to *ERG11* mutations, *ERG11* overexpression, alteration of membrane sterol composition, and increased expression of the efflux pump genes [14]. However, the frequency of each mechanism has been reported to vary among different demographic locations [15,16].

In Thailand, there are few data on the association between azole-resistant isolates of *C. tropicalis* and the underlying molecular mechanisms of azole resistance. We aimed to analyze the *ERG11* mutations and the expression of genes responsible for resistance among these isolates. The association between non-susceptible *C. tropicalis* isolates to fluconazole and risk factors for *C. tropicalis* blood infection were also investigated.

## 2. Materials and Methods

### 2.1. Blood Isolates of C. tropicalis

Blood isolates of *C. tropicalis* were collected from patients with *C. tropicalis* candidemia between January 2015 and December 2019 at Siriraj Hospital, a large tertiary center in Bangkok. Only the first strain isolated from the first episode of *C. tropicalis* was included in the study. All isolates were identified using a Microflex LT mass spectrometer (Bruker Daltonik, Bremen, Germany) according to the recommendations of the manufacturer. The study was approved by the Siriraj Institutional Review Board (certificate of approval Si 802/2019).

### 2.2. Definitions and Data Collection

We collected data from medical records including demographic data, comorbidities, predisposing factors, and clinical characteristics. Comorbidities were assessed using Charlson’s Comorbidity Index [17]. The source of candidemia was defined using the Infectious Diseases Society of America guidelines [18]. Predisposing factors on the day of bloodstream infection diagnosis included corticosteroid and antimicrobial administrations, parenteral nutrition, transplantation within 30 days, mechanical ventilation, placement of a central venous catheter at the time of onset, neutropenia (absolute neutrophil count ≤ 500 cells/mL). The severity of clinical presentation was evaluated using the Sequential Organ Failure Assessment Score (SOFA) [19]. Isolates from patients with either non-retrievable data or no antifungal susceptibility test results were excluded (Figure 1). The association between azole resistance and candidemia due to *C. tropicalis* was divided into two groups: the susceptible group (minimal inhibitory concentration {MIC} for fluconazole ≤ 2 μg/mL) and the non-susceptible group (MIC for fluconazole ≥ 4 μg/mL) [20,21].

### 2.3. Antifungal Susceptibility Test

All recovered *C. tropicalis* isolates were subjected to a retrospective antifungal susceptibility test (AFST) using Sensititre YeastOne (ThermoFisher, Waltham, MA, USA) which comprised nine drugs (fluconazole, voriconazole, itraconazole, posaconazole, 5-flucytosine, anidulafungin, micafungin, caspofungin, and amphotericin B). Interpretations of this colorimetric microdilution method followed the manufacturer’s recommendations. To ensure the commercial test’s reliability, all viable isolates used for the molecular studies were retested for fluconazole susceptibility by the broth microdilution method following the Clinical Laboratory Standards Institute (CLSI) documents M27A3 and M60 [21,22]. *Candida parapsilosis* ATCC 22019 and *Candida krusei* ATCC 6258 were used as quality controls. MICs by the commercial test were highly reproducible, with 97.7% of repeat testing results being within a single 2-fold dilution of the samples as shown in Appendix A.

### 2.4. Sequencing of the ERG11 Gene

Complete sequencing of the *ERG11* gene was performed on 53 isolates of the fluconazole-resistant group (one isolate was lost during sample collection) and 36 randomly selected isolates of the fluconazole susceptible group. DNA was extracted following the protocol described previously [23]. The amplified PCR products were sequenced using four primer pairs (Appendix A). The PCR protocol was used following the manufacturer’s recommendation for the Phusion Plus DNA Polymerase (ThermoFisher, Waltham, MA, USA) which recommended denaturation at 98 °C for 10 s, annealing at 60 °C for 10 s, and extension at 72 °C for 30 s. The amplified fragment was purified and directly used for bidirectional sequencing by Macrogen Inc. Korea using the PCR primers. The nucleotide sequences were examined for sequencing errors and heterozygous. The heterozygous was visually detected as the double peaks with strong intensity on the same locus and both strands and edited by Bionumeric software (version 8.0; Applied Maths NV, Sint-Martens-Latem, Belgium). The edited nucleotide sequences were compared with the *ERG11* of *C. tropicalis* MYA-3404 (GenBank accession number: XM_002550939).

### 2.5. Quantitative Real-Time RT-PCR of the ERG11, CDR1, and MDR1 Genes

Quantitative real-time RT-PCR was carried out on 80 isolates consisting of 47 isolates with fluconazole resistance (A total of 53 fluconazole-resistant isolates were excluded eight slow-grower isolates.) and 33 randomly selected isolates with fluconazole susceptibility. The *C. tropicalis* isolates were grown in the yeast extract peptone dextrose (YPD) broth and incubated at 37 °C in a shaker incubator at 200 rpm until the mid-log phase. Then, fluconazole at ¼ MIC level of each isolate was added to the media and continued incubation for 2 h which all isolates were grown to exponential phase. Meanwhile, the isolates were grown in the drug-free media under the same condition that was used as controls. After that, the isolates with- and without- the drug condition were extracted for total RNA [24]. Total RNA products were extracted using RNeasy Kits (Qiagen, Hilden, Germany) and reverse transcribed using the High-Capacity cDNA reverse transcription kit (ThermoFisher, Waltham, MA, USA). The cDNA products were obtained from the *ERG11*, *CDR1*, and *MDR1* genes using primers in Appendix A and measured with the LightCycler 480 instrument (Roche, Basel, Switzerland). The experiments were technical triplicated to ensure the reliability of the method. The expression of these genes was determined by the 2^−ΔΔCT^ method using the *ACT1* gene as an internal control [25]. The studied isolates were compared to the three genes’ expression during exposure to the drug relative to their expression’s growth in the drug-free condition.

### 2.6. Statistical Analysis

The minimal sample size for the prediction model of 137 patients was calculated using an azole resistance rate of 30% with 237 isolates over 5 years [8]. The categorical variables are presented as frequencies and compared using the chi-squared test or Fisher’s exact test. Continuous variables are shown as medians and interquartile ranges and analyzed with Student’s *t*-test and ANOVA. Univariate and multivariate regression analyses selected variables with a *p*-value < 0.2 and presented them with an odds ratio (OR) and 95% CI. The reliability and internal validation of the prediction model were using the rule of 10 events per covariate and bootstrapping, respectively [26]. All statistical analyses were performed with IBM SPSS Statistics for Windows (version 22.0; IBM, Armonk, NY, USA) and Prism software (version 9.4.1; GraphPad, San Diego, CA, USA).

## 3. Results

188 isolates of *C. tropicalis* bloodstream infections were included in the study. Forty-nine isolates were excluded, which consisted of 20 isolates with unretrievable data and 29 isolates with a lack of AFST results. The prevalence of non-susceptible fluconazole *C. tropicalis* was 40.4% (76 isolates) which consists of 54 fluconazole-resistant isolates and 22 fluconazole susceptible-dose dependent isolates.

### 3.1. Demographic Data and Clinical Characteristics

Age, admission to intensive care, Charlson’s comorbidity index, and severity at the time of onset were similar for both the fluconazole susceptible and resistant groups. Leukemia, lymphoma, recent chemotherapy administration, and exposure to carbapenem and azole were more common among patients with non-susceptible *C. tropicalis* blood isolates, as presented in Table 1.

### 3.2. Log Regression Analysis of Non-Susceptible Fluconazole C. tropicalis Blood Isolates

Independent factors associated with non-susceptible fluconazole *C. tropicalis* blood isolates were male sex (OR, 0.38; 95% CI, 0.19–0.77; *p* = 0.007); leukemia (OR, 3.15; 95% CI, 1.03–9.63; *p* = 0.044) and recent administration of azoles (OR, 10.56; 95% CI, 3.56–31.32; *p* = <0.001; Table 2).

### 3.3. Missense Mutation in the ERG11 Gene

The *ERG11* missense mutations were found in 44 of 53 resistant isolates (83.0%, Table 3) and absent in all susceptible strains. The most common base substitution was A395T (28 of 53, 52.8%) and the A395W substitution was the second most common (13 of 53, 24.5%). The A428G substitution was founded in two of 53 isolates (3.8%) and the A395W/T769C substitution was founded in only one isolate (1 of 53, 1.9%).

### 3.4. Expression Level of the ERG11, MDR1, and CDR1 Genes

The mean expression level in the *ERG11*, *MDR1*, and *CDR1* genes was 1.10, 1.05, and 1.16, respectively (all data shown in Appendix A). There were no statistically significant differences in the expression of the *ERG11*, *MDR1*, and *CDR1* genes between the azole-resistant and azole-susceptible groups (Figure 2). Interestingly, we found the azole-resistant isolates without *ERG11* mutations (R-WOM) had a significantly higher level *CDR1* expression (2.07) in the *CDR1* gene than the resistant isolates with *ERG11* mutations (R-WM, 1.01) and susceptible group (S, 1.13).

## 4. Discussion

This retrospective study revealed the prevalence of strains with dose-dependent fluconazole susceptibility and resistance was slightly higher than a previous report in 2013–2015, which found that the prevalence of dose-dependent susceptible and resistant strains was 9.5% to 11.7% and 20.2% to 28.7%, respectively [11].

Interestingly, the rate of recent exposure to azole was less (17%) than the rate of resistance to azole (28.7%). The earlier study reported that the azole-resistant group had a history of azole use in only one-third of the isolates [27]. Environmental spreading of resistant *C. tropicalis* strains was possible, according to previous reports on *C. tropicalis* and *C. parapsilosis* [6,28]. However, we have no direct evidence of patient-to-patient transmission, although the history of azole exposure was still a strong independent risk factor as in previous studies [12,13].

We report for the first time that leukemia and female sex are factors associated with *C. tropicalis* isolates resistant to azole. Leukemia increases exposure to antifungal prophylaxis to prevent invasive fungal disease after chemotherapy [29]. Men have lower rates of exposure to antifungal agents than women who may have chronic vulvovaginal candidiasis [30]. Therefore, we recommend the strict use of antifungal treatment according to the indications of the guidelines to reduce antifungal pressure [31,32].

The *ERG11* mutations found in our study consisted of four sites that were reported in previous studies [15,16,33,34]. Furthermore, the A395T mutation was the major missense mutation site found in more than 80% of previous studies and resulted in a high fluconazole MIC, especially as a homozygous mutation [15]. The A395T mutation was shown to alter the binding of heme to azole without altering enzyme activity which may be the reason for the isolates in our study that this mutation does not need the *ERG11* gene overexpression when counteracted with azoles [35]. However, further studies of the *UPC2* genes which are involved in the *ERG11* regulation are warranted [36].

Upregulation of the *MDR1* and *CDR1* genes may play a role in resistance to azoles, which was described in previous studies [15,16]. Similar to the previous studies, our results did not report a statistically significant difference in *MDR1* and *CDR1* gene expression between the azole-resistant and azole-susceptible isolates [15,37]. However, we found a higher level of expression of the *CDR1* gene in the R-WOM isolates. A recent study also revealed the role of the Tac1, which regulates the overexpression of the *CDR1* gene [38]. However, further investigations are needed to study the precise function of the other efflux pump genes such as *CDR2* and *CDR3* genes in azole-resistant *C. tropicalis* from our institute [37,38].

In conclusion, our results revealed two factors: female sex and leukemia that had not previously been reported in association with blood isolates of *C. tropicalis* resistant to azole. While *ERG11* missense mutations played an important role in azole resistance, neither higher expression of the *ERG11* nor drug transporter genes contributed to this phenotype. Other resistance mechanisms and additional genotyping to evaluate the potential the environmental transmission of the azole-resistant isolates remain to be further studied.

## Figures and Tables

**Figure 1 jof-08-00983-f001:**
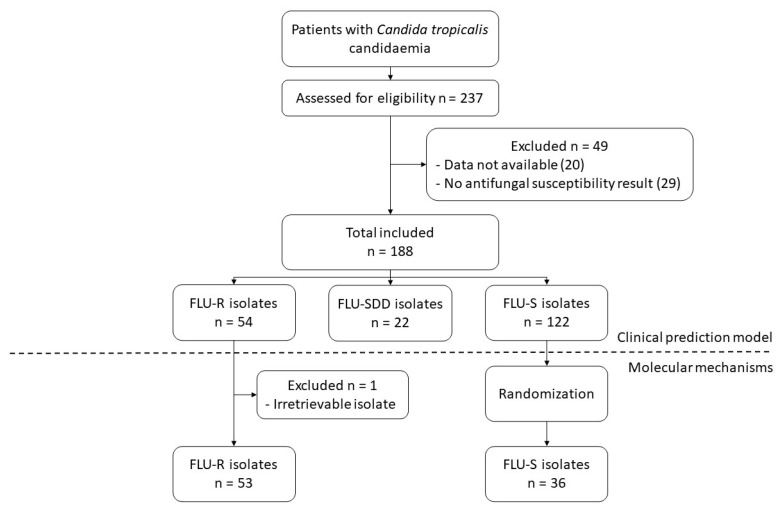
Flowchart of the study (Abbreviation: FLU-R, fluconazole-resistant; FLU-SDD, fluconazole susceptible-dose dependent; FLU-S, fluconazole-susceptible).

**Figure 2 jof-08-00983-f002:**
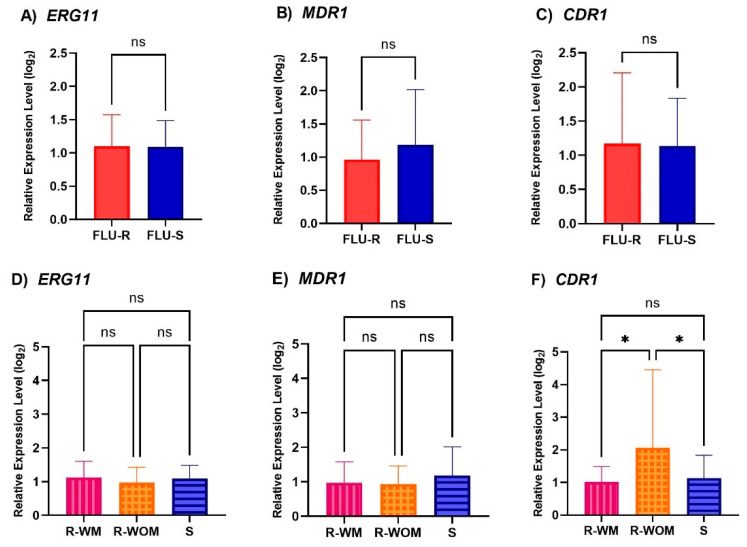
(**A**–**C**) The relative expression level of *ERG11*, *MDR1*, and *CDR1* genes of fluconazole resistant isolates; FLU-R, and fluconazole susceptible isolates; FLU-S. (**D**–**F**) The relative expression level of *ERG11*, *MDR1*, and *CDR1* genes comparisons among the resistant isolates with detected mutations; R-WM, the resistant strains without detected mutations; R-WOM, and the susceptible isolates; S. The results were performed in technical triplicates. Values are expressed as the mean ± standard deviation. * indicates statistically significant (*p* < 0.05) whereas ns means non-significant difference.

**Table 1 jof-08-00983-t001:** Demographic data and clinical characteristics of patients with *Candida tropicalis* bloodstream infection.

Characteristic	Total *N* = 188, *n* (%)	Fluconazole Non-Susceptible Isolates *n* = 76, *n* (%)	Fluconazole Susceptible Isolates *n* = 112, *n* (%)	*p* Value
Demographic data				
Age, years ± SD	58.0 ± 23.6	57.9 ± 21.7	58.1 ± 25.0	0.969
Male sex	88 (46.8)	28 (36.8)	60 (53.6)	0.024
Intensive care unit admission	87 (46.3)	32 (42.1)	55 (49.1)	0.345
Comorbidities				
Myocardial infarction	23 (12.2)	8 (10.5)	15 (13.4)	0.556
Cerebrovascular disease	15 (18.0)	3 (3.9)	12 (10.7)	0.093
Chronic liver disease	25 (13.3)	8 (10.5)	17 (15.2)	0.357
Diabetes mellitus	47 (25.0)	16 (21.1)	31 (27.7)	0.303
Chronic kidney disease	20 (10.6)	6 (7.9)	14 (12.5)	0.315
Solid tumor	29 (15.4)	10 (13.2)	19 (17.0)	0.478
Leukemia	21 (11.2)	13 (17.1)	8 (7.1)	0.033
Lymphoma	37 (19.7)	20 (26.3)	17 (15.2)	0.059
CCI ^1^, mean ± SD	4.5 ± 2.6	4.5 ± 2.7	4.5 ± 2.6	0.963
Predisposing factors ^2^				
Corticosteroid administration	51 (27.1)	22 (28.9)	29 (25.9)	0.644
Chemotherapy administration	49 (26.1)	26 (34.2)	23 (20.5)	0.036
Parenteral nutrition	91 (48.4)	34 (44.7)	57 (50.9)	0.407
Transplantation	6 (3.2)	4 (5.3)	2 (1.8)	0.224
Antibacterial agents use ^2^				
Penicillin	114 (60.6)	47 (61.8)	67 (59.8)	0.781
Cephalosporin	71 (37.8)	25 (32.9)	46 (41.1)	0.256
Carbapenem	153 (81.4)	67 (88.2)	86 (76.8)	0.049
Fluoroquinolone	76 (40.4)	32 (42.1)	44 (39.3)	0.699
Metronidazole	38 (20.2)	17 (22.4)	21 (18.8)	0.544
Glycopeptide	90 (47.9)	40 (52.6)	50 (44.6)	0.282
Colistin	46 (24.5)	18 (23.7)	28 (25.0)	0.837
Antifungal agents use ^2^				
Echinocandins	9 (4.8)	3 (3.9)	6 (5.4)	0.741
Amphotericin B	6 (3.2)	4 (5.3)	2 (1.8)	0.224
Azoles	32 (17.0)	25 (32.9)	7 (6.3)	<0.001
Severity at onset				
Sepsis	169 (89.9)	69 (90.8)	100 (89.3)	0.737
SOFA score, mean ± SD	7.7 ± 4.6	7.5 ± 4.4	7.8 ± 4.8	0.570
Septic shock	60 (31.9)	21 (27.6)	39 (34.8)	0.299
Neutropenia	49 (26.1)	25 (32.9)	24 (21.4)	0.079
Mechanical ventilation use	106 (56.4)	40 (52.6)	66 (58.9)	0.393
CVC ^3^ in place	111 (59.0)	47 (61.8)	64 (57.1)	0.520
Source				
Primary	110 (58.5)	46 (60.5)	64 (57.1)	0.644
CRBSI ^4^	68 (36.2)	27 (35.5)	41 (36.6)	0.880
Abdomen	3 (1.6)	2 (2.6)	1 (0.9)	0.567

^1^ CCI, Charlson’s comorbidity index. ^2^ Within 30 days before the onset of candidemia. ^3^ CVC, central venous catheter. ^4^ CRBSI, catheter-related bloodstream infection.

**Table 2 jof-08-00983-t002:** Univariate and multivariate regression analyses of factors associated with fluconazole non-susceptible *C. tropicalis* blood isolates.

Factors	Univariate Analysis			Multivariate Analysis		
	Crude OR ^1^	95% CI	*p*-Value	Adjusted OR ^1^	95% CI	*p*-Value
Male sex	0.51	0.28–0.92	0.025	0.38	0.19–0.77	0.007
Cerebrovascular disease	0.11	0.09–1.26	0.106	0.22	0.04–1.12	0.068
Leukemia	2.68	1.05–6.83	0.039	3.15	1.03–9.63	0.044
Lymphoma	2.00	0.97–4.13	0.062	1.77	0.77–4.06	0.177
Recent chemotherapy administration ^2^	2.01	1.04–3.89	0.038			
Recent carbapenems use ^2^	2.25	0.99–5.12	0.053	2.44	0.94–6.35	0.068
Recent azoles use ^2^	7.35	2.98–18.13	<0.001	10.56	3.56–31.32	<0.001
Neutropenia at onset	1.80	0.93–3.47	0.081			

^1^ OR, odds ratio. ^2^ Within 30 days before the onset of candidemia.

**Table 3 jof-08-00983-t003:** Missense mutations in the *ERG11* gene in resistant strains.

Amino Acid Substitutions	Nucleotide Mutation	Number of Isolates (%)	MIC of Fluconazole (mg/L)		
			Range	MIC_50_	MIC_90_
Y132F, S154F	A395T, A461T	28 (52.8%)	32 to >256	>256	>256
	A395W ^1^, A461Y	13 (24.5%)	8 to >256	32	128
Y132F, S154F, Y257H	A395W, A461Y, T769Y	1 (1.9%)	128–128	128	128
K143R	A428G	2 (3.8%)	64–64	64	64

^1^ Underline letters represent heterozygous mutation.

## Data Availability

The data presented in this study are available on request from the corresponding author. The data are not publicly available due to the Thailand Personal Data Protection Act.

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
