# Peer review of "The Risk Factors and Mechanisms of Azole Resistance of Candida tropicalis Blood Isolates in Thailand: A Retrospective Cohort Study"

_jof, 2022, doi:10.3390/jof8100983_

Round 1
Reviewer 1 Report
Congrats to the authors for completing this work. Here are my comments for further improvement -
General comment: There is a lack of novel findings in the work. The analysis and findings seem more suitable for a Brief Communication rather than an Original Article.
1. The Introduction section has references >5 years old. Please include most recent studies
2. "Furthermore, the prevalence of azole resistance is a positive trend in 30 Thailand and China [6,7]." - Please elaborate whether this observation applies only for Candida tropicalis or all Candida species? Also mention the trend seen in other Asian countries.
3. "The prevalence of fluconazole C. tropicalis isolates that are not susceptible to fluconazole in...." - meaning unclear, needs rephrasing
4. "The azole resistance of C. tropicalis was described in four major mechanisms comprised of ERG11..." - needs rephrasing
5. "Patients who had irrecoverable data and 62 did not take an antifungal susceptibility test were excluded. " - needs rephrasing...
6. Methodology: The CLSI M60 is the guideline for conducting and interpreting a reference method using the broth microdilution method performed in-house. It is recommended that the isolates be tested with the broth microdilution method to compare results of commercial tests like the Sensititre YeastOne for robustness.
7. Elaborate as to what method was used for gene sequencing with the primers
8. "The heterozygous was visually detected with strong overlapping peaks..." - needs rephrasing as meaning is unclear
9. "The prevalence of nonsusceptible fluconazole C. tropicalis was 40.4% (76 isolates). They comprised 54 episodes of fluconazole-resistant isolates and 22 episodes of fluconazole-susceptible independent isolates." - meaning is unclear
10. Clinical characteristics - please check if SOFA scores have been included in the results and analyzed, as mentioned in the methods section
11. What is the potential clinical/therapeutic clinical significance of these missense mutations? Mention in discussion
12. First sentence in Discussion - Is reference 8 related to the present study?
13. "Men have lower rates of exposure to antifungal agents than women who may have chronic vulvovaginal candidiasis [30]." - Did the women in this study have this history?
14. The discussion seems to suggest that because men are less exposed to antifungal agents prior to invasive Candida infection, their isolates must be azole susceptible but the study findings are to the contrary in the study. Am I missing something here?
15. "However, our results did not report statistical differences in the level of expression as in previous studies [23,24,35]." - What could be the possible reasons? Include the explanations in the Discussion
Reviewer 2 Report
I consider that the manuscript is well written and contains important resistance results specifically with Candida tropicalis.
The authors describe that they performed susceptibility test (AFST) using Sensititre YeastOne (Thermo Fisher Scientific Inc.), however, they do not show the susceptibility data, it would be important to add a table with these data or send it as supplementary material.
It is strongly recommended to use an ANOVA statistical test with Tukey's or Bonferroni correction postest that are in programs like PrismGraph Pad.
Reviewer 3 Report
In general, the article is well written, all results are performed and presented correctly. However, there are a few shortcomings:
Line 35: exposure of which antibacterial drug affect azole candida infection?
Line 46: How many isolates did you collect, and how many were taken to the research? In the following chapters, there are different no. of isolates. In methods 53 (resistant) and 36 (susceptible), in results 54 and 22? What is the reason of that?
Line 49: Please explain what it means: Only the first episode of C. tropicalis candidemia was included in the study. Maybe you should rephrase this sentence.
- Why do you use so many times word: episodes instead of samples, and isolates?
Table 2. Included information about all samples you collected? Maybe it will be better to present or emphasize this information for tested isolates?
The study of the level of the MDR and CDR1 genes may an effect on the method and time storage of isolates. This type of test should be performed as soon as possible after isolating the isolate from the patient. An experiment can be carried out by exposing a given strain to azoles and testing the expression level. Without exposure or long storage in the laboratory, such low expression levels could have been affected.
You wrote that you are conducting research for 8 other antimycotics why this data has not been presented?
Reviewer 4 Report
Dear authors,
I have evaluated your manusript “The risk factors and mechanisms of azole resistance of Candida 2
tropicalis blood isolates in Thailand; a retrospective cohort study “.
The increase in cases of azole resistance and evaluation of mechanisms of resistance in Candida tropicalis in Thailand presents interest to readers. The text should be carefully read and all the flaws corrected. The weak points are the lack of mechanisms of resistance in a part of studied isolates and the visual method for heterozygous detection.
Reviewer 5 Report
The study investigates azole resistance mechanisms of invasive C.tropicalis isolates from Thailand. The increase of C.tropicalis among candidemia causing agents and raising azol resistance were reported previously. Y132F was the most common mutation and no significant difference of ERG11, MDR1, and CDR1expression were found. Resistance mechanisms in all isolates could not be determined, which might lead to further studies. I think the study might benefit from some clarifications in the discussion section.
Line 146: "This retrospective study revealed a sharp increase in nonsusceptible fluconazole C. 146 tropicalis blood isolates from 0% in 2009 to 40.9% in 2019 [8]." Reference 8 tested only 8/41 of the C.tropicalis isolates. If possible, other studies form this center and/or from this area might be added. Also, a citation is needed for 2019 R rates.
Line 147: "Furthermore, the prevalence 147 of intermediate and resistant strains to fluconazole was slightly higher than reported in a 148 2015 study; from 9.5% to 11.7% and 20.2% to 28.7%, respectively [26]." When reporting resistance rates from different studies, the year the isolates were obtained should be stated instead of the year of publication. Here, reference was published in 2016 (not 2015, perhaps online access was in 2015), and the isolates were form 2013-2015.
Line 159: "Men have lower rates of exposure to antifungal agents than women who may have chronic vulvovaginal candidiasis". So, any suggestions on why males have higher fluconazol resistant C.tropicalis isolates despite potentially lower exposure to azoles?
Line 176: "Furthermore, the prevalence of azole-resistant isolates was higher than the usage of antifungal drugs, suggesting the possibility of environmental transmission in the healthcare setting." As this study did not perform genotyping, this might be stated as a possibility/further study option/shortcoming but not as a conclusion of this study.
